# Pharmacist-led educational interventions provided to healthcare providers to reduce medication errors: A systematic review and meta-analysis

Myriam Jaam[1‡]*, Lina Mohammad Naseralallah[1☉], Tarteel Ali Hussain[1☉], Shane Ashley Pawluk[2,3‡]

1 Clinical Pharmacy and Practice Department, College of Pharmacy, QU Health, Qatar University, Doha, Qatar, 2 Children's & Women's Health Centre of British Columbia, Department of Pharmacy, Vancouver, British Columbia, Canada, 3 Faculty of Pharmaceutical Sciences, University of British Columbia, Vancouver, British Columbia, Canada

☉ These authors contributed equally to this work.
‡ MJ and SAP are senior authors.
* myriamj@live.co.uk

**Data Availability Statement:** All relevant data are within the manuscript and its Supporting Information files.

## Abstract

### Introduction

Medication errors are avoidable events that can occur at any stage of the medication use process. They are widespread in healthcare systems and are linked to an increased risk of morbidity and mortality. Several strategies have been studied to reduce their occurrence including different types of pharmacy-based interventions. One of the main pharmacist-led interventions is educational programs, which seem to have promising benefits.

### Objective

To describe and compare various pharmacist-led educational interventions delivered to healthcare providers and to evaluate their impact qualitatively and quantitatively on medication error rates.

### Methods

A systematic review and meta-analysis was conducted through searching Cochrane Library, EBSCO, EMBASE, Medline and Google Scholar from inception to June 2020. Only interventional studies that reported medication error rate change after the intervention were included. Two independent authors worked through the data extraction and quality assessment using Crowe Critical Appraisal Tool (CCAT). Summary odds ratios (ORs) with 95% confidence intervals (CIs) were calculated using a random-effects model for rates of medication errors. Research protocol is available in The International Prospective Register of Systematic Reviews (PROSPERO) under the registration number CRD42019116465.

**Funding:** The author(s) received no specific funding for this work.

**Competing interests:** No authors have competing interests.

## Results

Twelve studies involving 115058 participants were included. The two main recipients of the educational interventions were nurses and resident physicians. Educational programs involved lectures, posters, practical teaching sessions, audit and feedback method and flash cards of high-risk abbreviations. All studies included educational sessions as part of their program, either alone or in combination with other approaches, and most studies used errors encountered before implementing the intervention to inform the content of these sessions. Educational programs led by a pharmacist were associated with significant reductions in the overall rate of medication errors occurrence (OR, 0.38; 95% CI, 0.22 to 0.65).

## Conclusion

Pharmacist-led educational interventions directed to healthcare providers are effective at reducing medication error rates. This review supports the implementation of pharmacist-led educational intervention aimed at reducing medication errors.

## Introduction

Despite improvements in the medical field seen through advanced technology and computerized systems, we still witness medication errors taking place. Medication errors are defined by the National Coordinating Council for Medication Error Reporting and Prevention (NCCMERP) as "any preventable event that may cause or lead to inappropriate medication use or patient harm while the medication is in the control of the health care professional, patient, or consumer" [1]. Numerous studies have reported that medication errors constitute a major reason for increased hospital stay, delayed medication administration, and an endless increase in medical costs [2–4]. In the United States 7,000 to 9,000 people die each year as a result of medication errors [5]. The global impact of medication errors as reported by the World Health Organization is estimated to be 42 billion USD each year [6]. The cost of a medication error can reach over $100 000 USD per patient [3].

The Majority of medication errors reported in the literature include prescribing errors requiring a pharmacist intervention [4, 7–9]. The reported prevalence of prescribing errors was reported to range from 2% to 94% [10]. These include among others, wrong drug, wrong dose, wrong route, wrong duration, and wrong formulation [7–11]. The results of a systematic review reporting error rates suggest that up to one prescribing error per patient takes place within a hospital setting [4]. Pharmacists are well-positioned to improve medication utilization and improve patient safety [12, 13].

Multiple interventions were tested for efficacy in reducing medication errors and studies have documented that pharmacists play a major role in reducing these medication errors [14–17]. Ward-based clinical pharmacists reduce medication errors by providing real-time advice to physicians rather than recommending changes after prescribing has occurred [18–20]. There is an important role of pharmacists in developing a medication safety strategy within their practice to reduce errors and improve the medication utilization process. One of the key elements of such a strategy is providing education to healthcare practitioners and prescribers since "too often, there is insufficient education of healthcare providers on medication safety topics" [13]. Educational interventions implemented by pharmacists within the pediatric ward

have been reported to increase detection and reporting of medication errors while at the same time, reducing the severity of medication errors [21].

Pharmacist-led educational interventions seem to show benefit in reducing medication errors; however, it is not clear how often these educational sessions should run, nor what content should be included to provide optimal benefit in reducing medication errors [22]. To the best of our knowledge, this is the first systematic review and meta-analysis aimed to investigate the effectiveness of pharmacist-led educational interventions provided to healthcare providers to reduce medication errors both quantitatively and qualitatively.

## Materials and methods

This systematic review and meta-analysis followed the Preferred Reporting Items for Systematic Reviews and Meta-Analyses (PRISMA) reporting methodology [23]. The research protocol is available in The International Prospective Register of Systematic Reviews (PROSPERO) under the registration number CRD42019116465 [24].

### Database and search strategy

Studies were captured from five databases: Cochrane Library, EBSCO, EMBASE, Medline, and Google Scholar, from inception until June 2020. In addition to databases, references of studies were also manually screened for relevant articles. To identify relevant studies, free text and MeSH terms were used and combined with Boolean operators 'AND' to combine terms of different categories and 'OR' to combine terms within one category. Table 1 summarizes the terms used for the search strategy. The limits applied include searching within title and abstract and using the English language in reporting the article. The search strategy was applied by two researchers, independently, to ensure consistency. Database hits and references identified from reference screening were then transferred to an excel sheet for removing duplicates. Screening of titles and abstracts was then completed by two independent authors. Any discrepancy was resolved through full-text screening of articles.

### Eligibility criteria

Articles were included if they (i) were published in English; (ii) investigated pharmacist-led educational interventions provided to healthcare providers, and (iii) reported medication error rate or number before and after the intervention.

Articles were excluded if they were (i) non-interventional descriptive studies, (ii) systematic reviews or meta-analyses, (iii) investigating pharmacy reconciliation and their effect on medication discrepancies, or (iv) were led by students or pharmacy technicians. Additionally, editorials, opinions, abstract-only studies were excluded.

**Table 1. Search terms.**

| Category | Search terms |
|---|---|
| Category A | Pharma* |
| Category B | Medication errors OR Adverse effect OR mistake OR inappropriate prescribing OR safe prescribing OR medication discrepancy OR medication safety OR adverse drug event |
| Category C | Prevent* OR Reduc* OR decrease OR augment OR effectiveness OR improvement |
| Category D | Education OR training OR teach |

## Data extraction and quality assessment

A data extraction sheet was created and piloted on two randomly chosen included articles. The final data extraction form included: title, authors, year of publication, country, setting, objective, study type, medication error definition, outcome measure, the intervention provided, description of education provided, time frame, error rate -before and after-, conclusion, limitations and quality score. Whenever a study included multiple interventions, results of the pharmacist-led educational intervention alone were pooled out separately. The process was conducted by two independent researchers and consensus was sought through discussion.

The Crowe Critical Appraisal Tool (CCAT) was used for the quality assessment of included articles [25]. The CCAT covers eight domains: preliminary, introduction, design, sampling, data collection, ethical matters, results, and discussion. Each domain is scored out of five providing an overall score of 40 points which equates to a 100% score. The higher the score the better the quality of the article. Quality assessment was conducted by two independent researchers and the average overall quality score was used for reporting. The CCAT was previously used by the research team and high interrater reliability was demonstrated in previous research [26].

## Data synthesis and analysis

Meta-analysis was performed using the Mantel-Haenszel odds ratio (OR) with a 95% confidence interval. A random-effect model was used due to differences in study locations, the population covered, and study interventions. P-value was also calculated to reflect the overall test effect and a value of less than 0.05 was considered to be significant. Cochrane Collaboration Review Manager (RevMan) version 5.3 was used for the analysis.

## Results

### Identification and selection of studies

A total of 9354 articles were identified from the different databases and reference screening. After duplicate studies were removed (n = 6857), the remaining articles were screened by title and abstract by two researchers. A total of 12 articles were included in the systematic review and meta-analysis Fig 1.

### Study characteristics

The characteristics of included studies are presented in Table 2. Studies captured were published from 2007 to 2019. The studies found were conducted in different countries including Egypt (n = 3) [27–29], Australia (n = 2) [30, 31], USA (n = 2) [32, 33] and one each from Pakistan [34], Spain [35], Netherlands [36], Saudi Arabia [37] and Vietnam [38]. The majority of studies included looked at the pediatric population and covered different hospital units including neonatal/pediatric intensive care units (n = 3) [27, 35, 36], children ward [27], and one study was conducted in the pediatric surgery department [28]. Three studies covered multiple hospital departments [30, 32, 38] and three covered emergency/acute care hospital wards [29, 31, 37], while only one study was conducted in an outpatient clinic [33]. Medication error definition varied across the included studies; nonetheless, six studies covered prescribing errors which had a similar definition among the studies and included wrong drug, wrong dose, wrong frequency, wrong concentration, wrong or missed rate of administration, wrong or missed instructions for administration, unclear order or incomplete order [27, 28, 30, 32, 33, 35]. Additionally, medication errors included the use of error-prone abbreviations in prescribing in seven studies [27, 30–33, 35, 37], Other medication errors captured by the studies were

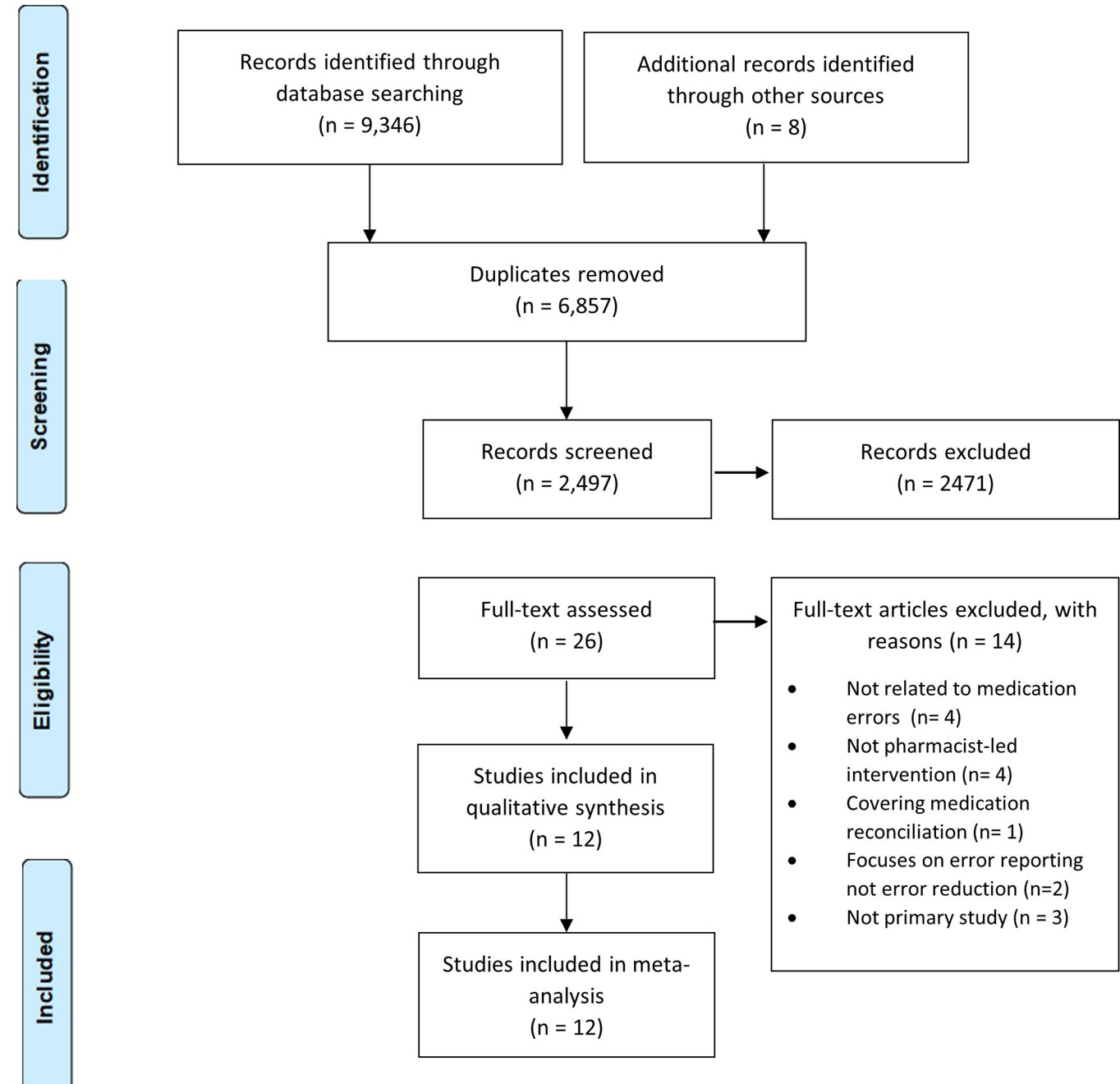

**Fig 1. Prisma flow diagram.**

administration and preparation errors (n = 4) [29, 34, 36, 38] and medication error occurring at any stage [28].

## Quality of included studies

Two raters with previous experience with using the CCAT [26] appraised each of the 12 studies, which resulted in 24 independent CCAT evaluations. The average score was reported (Table 3). The total score ranged from 56.25% to 92.5% and the overall assessment mean for all

**Table 2. Study characteristics.**

| Author/year (Reference) | Country/Setting | Targeted medication error | Baseline measurement duration | Baseline medication error rate | Education provided | Duration of intervention | Education frequency | Participants | Medication error rate post-intervention (duration) |
|---|---|---|---|---|---|---|---|---|---|
| Ahmed et al. (2017) [27] | Pakistan/Children ward in a teaching hospital | Administration errors | 2 months | 82.13% | • Lectures (number, frequency and duration NR). <br> • Preintervention errors informed content <br> • Covered: the types of medication administration errors made previously (based on patient charts) and reasons for these errors. <br> • A printed copy of the lecture was provided. | 1 month | NI | Nurses | 52.74% (2 months) |
| Alagha et al. (2011) [20] | Egypt/ PICU | Prescribing errors | 5 months | 78.10% | • Lectures (number and duration NR) <br> • Preintervention errors informed content <br> • Covered: Good prescribing and prescription writing, introducing new order chart, medication errors and their consequences, and orientation to the drug use assists and their use. <br> • A printed copy of the lectures was provided. <br> • A dosing sheet for the commonly used drugs (IV, oral, inhalation) was provided to residents and a copy was made available in all patient files. <br> • All residents received individualized written reports about their errors throughout the post-intervention phase. | NI | Once | New resident physicians | 35.20% (5 months) |
| Campino et al. (2009) [28] | Spain/ NICU | Prescribing errors | 5 months | 20.70% | • 15 informative sessions (duration NR) <br> • Covered: medication errors, a non-punitive culture on patient safety, and study aim. | 4 months | Fifteen | All health professionals | 3.00% (2 months) |
| Chedoe et al. (2012) [29] | Netherlands/ NICU | Preparation and administration errors | 1 month | 49.00% | • Theoretical teaching session (five, 1 hour each) <br> • Covered: calculation, reconstitution, compatibilities, administration rate, and aseptic technique of drug preparation and administration <br> • Individual practical teaching session (once for around 30 minutes) <br> • Covered: preparation and administration of all common ward medication and a short guided tour around the pharmacy department. <br> • Teaching aids: a PowerPoint presentation, a video presentation, made available on the hospital's intranet. <br> • A poster with recommendations for safe preparation and administration was placed in the preparation area. All guidelines outlining drug preparation and administration were updated and available on the ward. | 26 months | Six (5 lectures, 1 practical session) | Nurses | 31.00% (1 month) |

*(Continued)*

Table 2. (Continued)

| Author/year (Reference) | Country/Setting | Targeted medication error | Baseline measurement duration | Baseline medication error rate | Education provided | Duration of intervention | Education frequency | Participants | Medication error rate post-intervention (duration) |
|---|---|---|---|---|---|---|---|---|---|
| Fawaz et al. (2017) [21] | Egypt/ Pediatric surgery department in a teaching hospital | Medication errors including all stages (Prescribing, transcribing, Administration) categorized according to the NCCMERP | NI | 33.33% | • One educational session (duration NR)<br><br>• Covered: definition of medication errors, medication use process, the difference between medication errors and adverse drug events, barriers to reporting medication errors, categories of medication errors according to the NCCMERP, types of medication errors, USP medication errors reporting program, prevention strategies, and the role of clinical pharmacist in the operating room. | NI | Once | Physicians (pediatric surgery and anesthesia residents) | 32.32%<br><br>(NI) |
| Gursanscky et al. (2018) [23] | Australia/ Four general medical units in a tertiary hospital | Prescribing errors | 3 weeks | 58.00% | • Three 10-min educational sessions per week<br><br>• Covered: frequency, type, and severity of prescribing errors. The pharmacist discussed why the errors were considered incorrect or unsafe, the potential consequences of such errors, and the correct way to prescribe. Any questions or ambiguities were addressed. Subsequent sessions addressed the next most common and serious errors, usually covering two to three error types per session.<br><br>• Preintervention errors informed content using data from the 1–2 days immediately preceding each session.<br><br>• A printed copy of the lecture was provided. | 4 weeks | 3 sessions / week | Junior doctors | 37.00%<br><br>(1 month) |
| Haseeb et al. (2016) [30] | Saudi Arabia/ Acute care hospital | Use HRA | 1 month | 53.60% | • A series of interactive lectures (number, frequency, and duration NR)<br><br>• Covered: basic concepts of prescribing and dispensing errors, negative consequences of using HRAs in an emergency setting, and error prevention based on the Institute for Safe Medical Practices and the US Food and Drug Administration<br><br>• A printed list of abbreviations was inserted into medical records/patient charts, placed next to all hospital computers, and posted in the patient care area<br><br>• Pocket-sized flashcards with the HRA list provided for all staff<br><br>• Patient safety posters given to the hospital wards explaining HRA | 1 month | NI | All health professionals (i.e. doctors, nurses, pharmacists, and relevant assistants) | 25.50%<br><br>(1 month) |

(Continued)

**Table 2.** (Continued)

| Author/ year (Reference) | Country/ Setting | Targeted medication error | Baseline measurement duration | Baseline medication error rate | Education provided | Duration of intervention | Education frequency | Participants | Medication error rate post-intervention (duration) |
|---|---|---|---|---|---|---|---|---|---|
| Mostafa et al. (2019) [22] | Egypt/ Emergency hospital | Administration and preparation error categorized according to the NCCMERP | NI | 34.20% | • Lectures (number, frequency, and duration NR)<br>• Covered: medication safety, general advice about administering medications, as was a list of all the medications available in the emergency hospital, with a brief explanation of the proper administration techniques and storage precautions.<br>• Posters of medications and their proper administration were hung in all emergency departments and the emergency pharmacy. of<br>• Lectures and posters were in the Arabic language to make it easier for nurses to understand. | NI | NI | Nurses | 15.30% (NI) |
| Nguyen et al. (2013) [31] | Vietnam/ ICU and PSU | IV medications preparation and administration errors | 1 week | 64.00% | • Training program including classroom lectures: two 30-min teaching sessions with PowerPoint presentation<br>• covered: reconstitution, compatibility, administration rate, and drug preparation and administration techniques<br>• Practice-based education: one 45-min practical session covering the preparation and administration of commonly used medications and including discussion of patient cases<br>• Two posters on recommended practice for safe preparation and administration and emphasizing the adverse consequences of inappropriate procedures were attached to the wall of the preparation area.<br>• Written guidelines on the preparation and administration of commonly used IV drugs were made available on the ward. | 2 weeks | Three (2 lectures, 1 practical session) | Nurses | 48.90% (1 week) |
| Peeters et al. (2009) [25] | USA/ 13 hospital units | Prescribing errors | 1 month | 2.25% | • Divided into two phases: Phase 1—an hour-long lecture<br>• covered: definitions and categories of medical errors and medication errors, Institute of Medicine reports, Joint Commission medication management requirements, and institutional medication order policies<br>• Phase 2: a series of short (15 minutes), follow-up discussion sessions<br>• covered: prescribing errors identified<br>• Preintervention errors informed content | 6 months | Phase 1: once<br>• Phase 2: biweekly for 2 months then monthly for 4 months | Internal medicine residents | After phase 1 and early into phase 2: 1.51%<br>Post-intervention: 2.33% (1 month) |

(*Continued*)

**Table 2.** (Continued)

| Author/ year (Reference) | Country/ Setting | Targeted medication error | Baseline measurement duration | Baseline medication error rate | Education provided | Duration of intervention | Education frequency | Participants | Medication error rate post-intervention (duration) |
|---|---|---|---|---|---|---|---|---|---|
| Taylor et al. (2007) [24] | Australia/ ED | Error-prone prescribing abbreviations | 1 week | 31.80% | • Educational sessions (duration NR) in the form of small-group or one-on-one tutorials<br>• Distribution of name badge sized cards summarizing the information<br>• Placement of posters in the ED | 5 months | NI | Prescribing physicians and nurses | 18.70% (1 week) |
| Winder et al. (2015) [26] | USA/ Outpatient academic family medicine clinic | Prescribing error | 1 month | 18.60% | • The first intervention (optional) was one educational outreach visit delivered over 1 hour<br>• Covered: prescribing errors, applicable use of tools available in the EMR, drug information resources available within the EMR, and necessary elements for writing a prescription, unavailable medications, unapproved abbreviations, new medication labeling, the prescription review process, and questions using an audience response system as well as practice calculations for pediatric prescriptions.<br>• Preintervention errors informed content<br>• The second intervention involved the implementation of an audit and feedback method and a weekly newsletter.<br>• All residents received individualized written reports about their errors via EMR alerts. Errors found were reported to the institution's risk management department.<br>• Verbal feedback was provided as needed to prescribers based upon the level of severity of the error.<br>• Additionally, over 3 months following the educational outreach, all residents were sent a weekly newsletter of common errors via the family medicine email list. Weekly newsletters included contact information for clinical pharmacists, common errors identified during the previous week, error type, examples of the error, recommendations for correction, and highlights from the educational outreach. | First intervention: once. Second intervention: 3 months | First intervention: once. Second intervention: weekly | Resident physicians | 14.50% (1 week) |

PICU: Pediatric Intensive Care Unit. NICU: Neonatal Intensive Care Unit. PSU: Post-Surgical Unit. ED: Emergency Department. IV: Intravenous. NI: Not indicated. NR: Not reported. NCCMERP: The National Coordinating Council for Medication Error Reporting and Prevention. HRA: High-Risk Abbreviation. EMR: Electronic Medical Record

**Table 3. Cumulative quality assessment results.**

| Study | Preliminary | Introduction | Design | Sampling | Data analysis | Ethics | Results | Discussion | Overall Out of 40 (%) |
|---|---|---|---|---|---|---|---|---|---|
| Ahmed et al [27] | 3.00 | 3.00 | 3.00 | 1.50 | 3.50 | 3.00 | 3.00 | 2.50 | 22.50 |
| | | | | | | | | | (56.25%) |
| Alagha et al [20] | 5.00 | 5.00 | 3.00 | 0.00 | 2.00 | 0.00 | 5.00 | 5.00 | 25.00 |
| | | | | | | | | | (62.50%) |
| Campino et al [28] | 5.00 | 4.00 | 5.00 | 3.00 | 4.00 | 1.00 | 3.00 | 3.00 | 28.00 |
| | | | | | | | | | (70.00%) |
| Chedoe et al [29] | 5.00 | 5.00 | 4.00 | 4.00 | 5.00 | 4.00 | 5.00 | 5.00 | 37.00 |
| | | | | | | | | | (92.50%) |
| Fawaz et al [21] | 4.00 | 2.00 | 3.50 | 2.50 | 3.50 | 1.50 | 3.50 | 3.00 | 23.50 |
| | | | | | | | | | (58.75%) |
| Gursanscky et al [23] | 5.00 | 5.00 | 5.00 | 5.00 | 4.00 | 1.00 | 4.00 | 5.00 | 34.00 |
| | | | | | | | | | (85.00%) |
| Haseeb et al [30] | 5.00 | 5.00 | 5.00 | 5.00 | 4.00 | 2.00 | 4.00 | 5.00 | 35.00 |
| | | | | | | | | | (87.50%) |
| Mostafa et al [22] | 4.00 | 4.50 | 3.00 | 1.00 | 3.50 | 4.50 | 3.50 | 3.50 | 27.50 |
| | | | | | | | | | (68.75%) |
| Nguyen et al [31] | 5.00 | 5.00 | 5.00 | 3.00 | 4.00 | 4.00 | 4.00 | 5.00 | 35.00 |
| | | | | | | | | | (87.50%) |
| Peeters et al [25] | 4.00 | 5.00 | 4.00 | 5.00 | 4.00 | 2.00 | 2.00 | 4.00 | 30.00 |
| | | | | | | | | | (75.00%) |
| Taylor et al [24] | 3.00 | 5.00 | 2.00 | 4.00 | 3.00 | 4.00 | 3.00 | 2.00 | 26.00 |
| | | | | | | | | | (65.00%) |
| Winder et al [26] | 5.00 | 4.50 | 4.00 | 4.00 | 4.00 | 5.00 | 4.50 | 4.00 | 35.00 |
| | | | | | | | | | (87.50%) |
| **Overall** | 4.42 | 4.42 | 3.88 | 3.17 | 3.71 | 2.67 | 3.71 | 3.92 | 29.88 |
| | | | | | | | | | (74.68%) |

studies was 74.68%. Within the CCAT sections, the highest scores were for preliminary and introduction (4.42/5), while the lowest were for sampling (3.17/5) and ethics (2.67/5) (Table 3).

## Description of educational interventions

The 12 studies included in this review involved pharmacist-led educational interventions that varied in content, recipient group, duration, and frequency of delivery. The educational interventions targeted nurses (n = 4) with a focus on medication preparation and administration errors [29, 34, 36, 38] and resident physicians (n = 4) which focused on prescribing errors [27, 28, 32, 33]. The period of the interventions covered a wide range of duration with the longest being 26 months [36] and the shortest was conducted over two weeks [38], however, the investigators within three studies did not report intervention duration [27–29].

All included studies delivered informative didactic lectures as part of their intervention. This was a sole intervention in six studies [27, 28, 30, 32, 34, 35] or was combined with other forms of educational interventions, including posters (n = 5) [29, 31, 36–38]; practical teaching sessions [36, 38]; audit and feedback method along with a weekly newsletter [33]; and pocket-sized flashcards of high-risk abbreviations [37]. Studies that included posters in addition to lectures indicated that the posters were placed in the ward and included important recommendations and guidance for healthcare providers. On the other hand, studies that included practical

teaching sessions covered preparation and administration of commonly used medications in addition to individualized 30-minute sessions delivered to new residents or 45-minutes group discussions of patient cases [36, 38].

The frequency of the educational sessions varied from once only in two studies [27, 28] to a total of 21 sessions over a period of six months [32] while it was not reported in four studies [29, 31, 34, 37]. All studies did not report the number and duration of lectures, except for Chedoe et al (five 1-hour sessions) [36]; Gursanscky et al (twelve 10-minutes sessions) [30]; Nguyen et al (two 30-minutes sessions) [38]; Peeters et al (phase 1: one 1-hour session, phase 2: eight 15-minutes sessions) [32]; and Winder et al (one 1-hour session) [33]. Five out of the twelve studies distributed printed handouts that summarized the lecture content [27, 30, 31, 34, 37].

Different teaching aids were used during the lectures, such as PowerPoint presentation slides [36, 38], video presentation [36], or audience response system [33]. Seven studies used errors encountered before the sessions or error rates captured at the pre-intervention phase to inform the content of these sessions [27, 29–31, 33, 34, 38]. The educational sessions mostly covered aspects such as definition, type, causes, consequences, and prevention methods. Studies that focused on prescribing errors covered good prescription writing including the use of appropriate abbreviations within their educational sessions, while studies that focused on administration and preparation errors covered more technical teaching such as calculations, reconstitution, compatibilities, and drug administration rate. The number of attendees at educational sessions was not clear in all studies, and no description was provided on how attendees were informed about the educational sessions (Table 2).

## Effect of educational interventions

All studies included in the systematic review were eligible for meta-analysis (Fig 2). Ten of twelve studies indicated a clear and significant decrease in the incidence of medication errors after implementing a pharmacist-led educational intervention [27, 29–31, 33–38]. The pooled OR (n = 115 058 participants) across all studies was 0.38 (95% CI 0.22 to 0.65) (P = 0.0004). However, the results of these studies were substantially heterogeneous (Fig 2). Only two studies [28, 32] conducted in the USA and Egypt showed no significant difference in the occurrence of medication errors after applying the intervention. The study conducted in the USA had a very low rate of medication errors relative to other included studies and was divided into two phases, in which a significant reduction was observed during the intervention, but was not maintained one month post-intervention [32]. The educational intervention for the study conducted in Egypt consisted of one educational session, without follow-up sessions or distributing a printed copy of the lecture, and it showed a nonsignificant reduction of medication errors after the study intervention [28].

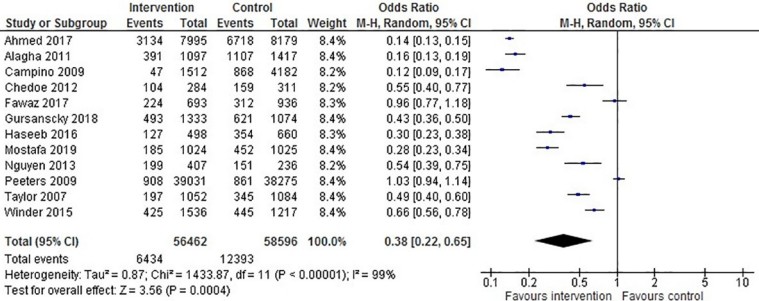

**Fig 2. Random effect model of pharmacist-led education and its association with medication errors.**

## Discussion

Majority of pharmacist-led interventions—targeting medication errors—reported in the literature focus on medication reconciliation or education provided to patients rather than healthcare providers [39–41]. This paper has shown an overall better outcome in favor of pharmacist-led educational interventions provided to healthcare providers on the rate of medication error occurrence.

Although most included studies showed improvement, two papers did not show a significant difference after implementing the intervention. The educational intervention in the first study conducted by Fawaz *et al* consisted of one lecture delivered to physicians without providing any printout or posters to summarize the content of the lecture which can serve as a constant reminder to recipients [28]. In the second study which was conducted by Peeters *et al.*, pharmacists delivered frequent lectures to residents throughout six months, yet the benefit was only noted during the intervention period but was not maintained one month post-intervention [32]. This could be due to the initial very low rate of medication errors relative to other included studies in this review. Additionally, similar to Fawaz *et al.* study, the study by Peeters *et al.* did not provide any form of constant reminder (e.g. printouts or posters) as part of the educational intervention. Multiple studies showed that various educational strategies, such as brochures or training activities, have beneficial outcomes including improving different aspects of knowledge and skills of healthcare workers [42–44]. This benefit is also highlighted within the four studies that used printed handouts which resulted in significant reduction in medication error rates, including the trial conducted by Alagha *et al.*, which used only one educational session [27, 30, 31, 34, 37]. In addition to the printed materials, Alagha *et al.* and Winder *et al.* provided individualized reports (written and through Electronic Medical Record (EMR) respectively) to the resident physicians about their own errors. This approach might have contributed to the significant reduction in medication errors in these studies because each resident would avoid the factors that lead to their prescribing errors; hence minimize them [27, 33].

The most intensive intervention was the one conducted by Winder *et al.* [33]. Within this study, researchers implemented multi-faceted interventions to reduce prescribing errors. This included four phases: a pre-intervention error assessment; followed by an optional educational intervention delivered to resident physicians; and error notification intervention through an audit and direct feedback to residents electronically and verbally depending on the error severity. This phase also included sending weekly newsletters of common errors over a period of three months and post-intervention error assessment conducted after one year. Within this study, the pediatric dosing errors were 36% lower among residents who attended the educational sessions compared to those who did not. This approach highlights the importance of continued audit and direct feedback to improve patient safety to be conducted along with healthcare practitioners' education. The importance of continuous electronic audit and feedback was, similarly, emphasized upon by a study conducted in the UK which covered over 40 general practices [45].

Additionally, the frequency and duration of educational intervention can play a role in maintaining good outcomes. It was noted that studies that used more frequent educational sessions such as the one reported by Campino and Gursanscky providing 15 and 12 sessions within four months and one month respectively, had better outcomes compared to other studies [30, 35]. Repeated encounters–as reported by Bruner–improves information retention [46]. It must be highlighted though that the mere repetition of the sessions might not be sufficient in improving practice. Brabeck *et al.* highlighted that deliberate practice significantly improves cognitive gain and brings about the motivation for more learning, hence, better performance

[47]. Therefore, the frequent educational sessions should always have a component of the application to have a better outcome.

The results of this meta-analysis highlight that many factors can lead to a significant reduction in medication errors post educational sessions. Medication errors can significantly reduce if pharmacists invest in providing education to healthcare providers that are supplemented with printed handouts summarizing the lecture content. Additionally, the provision of individualized medication error reports and the use of ward-based posters addressing preparation and administration errors can also help reduce prescribing and medication administration errors. For ease of access, pharmacists can also work on providing flashcards with instructions for good prescribing including high-risk abbreviations when addressing prescribing errors. Finally, having frequent application-based educational sessions can be of great benefit to healthcare providers.

Additionally, majority of the included studies used errors reported before the educational sessions to inform the content. Such an approach can contribute to the benefit observed in reduced medication errors. Based on the results of this systematic review, the authors recommend using frequent educational sessions supplemented with individualized prescriber reports to significantly minimize medication errors. The session content should be site-specific based on the types of medication errors encountered and summary material should be provided to attendees. Many articles emphasized the importance of continuous monitoring and reporting of errors to maintain a safe practice and reduce patient harm [48–50]. As such, these continuous reports should inform the content of the educational sessions.

The quality of reporting of the included studies was of an acceptable level. Nonetheless, the authors believe it can always be improved through the utilization of reporting tools. Within the CCAT, the item which scored the least was ethics which focuses on reporting of consent, ethics approval, and anonymity of data. This was followed by "sampling", which includes three components: sampling method, sample size, and sampling protocol. Sampling is a very important component that can affect the study power to detect a difference and can also affect the generalizability of the results [51]. Research on medication errors should maximize its quality through the use of reporting tools and checklists available in the literature and the Enhancing the QUAlity and Transparency of health Research (EQUATOR) Network [52].

Our results are consistent with previous systematic reviews and meta-analyses that studied the effectiveness of different clinical pharmacist interventions on medication error rates in adults and pediatric patients at diverse hospital departments [26, 41, 53–57]. The studies included multiple interventions such as reviewing orders, implementing a ward-based pharmacist, or designing educational programs. However, none of these papers focused on educational interventions or had a subgroup analysis to investigate their effectiveness in decreasing medication errors. Nonetheless, Manias *et al.* published a recent systematic review of all interventions to reduce medication errors in adult medical and surgical units which described similar outcomes to this systematic review. Pharmacist-led interventions were of great benefit and educational interventions to healthcare providers proved to be effective in reducing medication errors [58]. This emphasizes the importance of having drug experts take lead in delivering education to their colleagues.

Limitations within this study need to be acknowledged. Significant heterogeneity in the studies included in the meta-analysis was reported. This is due to major differences in the delivery of the educational sessions, variation in the settings, and differences in the type and definition of the targeted medication error. Additionally, almost all studies reported positive results with the pharmacist-led educational intervention and we cannot exclude the risk of publication biases. Moreover, the researchers within the included studies might have chosen the areas that are likely to show positive results since they included acute care, pediatric, and emergency settings. The off-label, unlicensed use of medicines, weight-related dose

adjustment, and other dosing calculations are widespread in these settings particularly in pediatrics, which may increase the risk of avoidable medication-associated harm. Therefore, they are more likely to report positive results. In addition, the setting for this review was mainly inpatient with only one study reporting findings in the outpatient setting. As a result, our findings are unlikely to be generalizable to all healthcare systems. Finally, the authors acknowledge that excluding non-interventional studies might have precluded many potentially good and relevant articles. However, the authors focused on interventional studies which are generally recognized as stronger evidence generating studies with less confounding [59].

Future investigations and randomized controlled studies are required to evaluate the impact of pharmacist-led educational intervention on other types of medication errors not covered in the studies included in this review, such as transcribing, dispensing, and monitoring errors. This will provide a broader picture of the capability of the pharmacists in providing effective education to healthcare providers that minimize all types of medication errors. The effect of pharmacist-led education to healthcare providers on medication errors in outpatient settings should gain more attention in further studies. This might include outpatient clinics, home care, and community pharmacies. The elderly population may also encounter special issues related to medication errors. For example, people living in care homes are often frail with multiple health conditions and take multiple medications. The administration of medication in this environment often differs to patients' own homes as it is provided by nursing staff or other personnel, thus raising particular issues around dispensing, administration, and monitoring problems, as well as staff training. Additionally, many pharmacist-led interventions target medication errors apart from providing education to healthcare providers, including providing medication reconciliation; including pharmacists inwards; and involving them in team discussions [39–41]. A study that compares the benefit of these interventions and determining priority intervention will be of significant benefit especially in areas where resources are limited. Medication errors are known to be linked to morbidity and mortality as highlighted by the well-cited report *"To Err is human"* [60], therefore, future studies should consider the outcome of morbidity and mortality as an extended outcome to medication error rates. This will provide a better understanding of the consequences of interventions done to tackle medication errors. Finally, the economic impact of this intervention and associated cost saving studies related to education can support resource allocation and improve patient safety.

## Conclusion

This systematic review and meta-analysis demonstrated that pharmacist-led educational interventions to healthcare providers are effective in reducing medication errors. The educational intervention was observed to be most effective when supplemented with printed handouts that summarize the session content, providing posters or pocket-sized flashcards to prescribers, and when healthcare providers receive an individualized written or electronic report about their medication errors. Additionally, frequent educational sessions were observed to be more effective than one-time sessions. These combined approaches serve as constant reminders to the healthcare practitioners to minimize medication errors at different stages of the medication-use process. Further research should consider the effect on morbidity, mortality related to medication errors and the economic impact of pharmacist-led education provided to healthcare providers.

## Supporting information

**S1 Checklist. PRISMA checklist.**
(DOC)

**S1 File. Search strategy.**
(DOCX)

## Author Contributions

**Conceptualization:** Myriam Jaam, Shane Ashley Pawluk.

**Data curation:** Myriam Jaam, Shane Ashley Pawluk.

**Formal analysis:** Myriam Jaam.

**Funding acquisition:** Myriam Jaam.

**Investigation:** Myriam Jaam, Lina Mohammad Naseralallah, Tarteel Ali Hussain, Shane Ashley Pawluk.

**Methodology:** Myriam Jaam, Lina Mohammad Naseralallah, Tarteel Ali Hussain, Shane Ashley Pawluk.

**Project administration:** Myriam Jaam, Shane Ashley Pawluk.

**Resources:** Myriam Jaam, Lina Mohammad Naseralallah, Tarteel Ali Hussain, Shane Ashley Pawluk.

**Software:** Myriam Jaam, Lina Mohammad Naseralallah, Tarteel Ali Hussain, Shane Ashley Pawluk.

**Supervision:** Myriam Jaam.

**Validation:** Myriam Jaam.

**Visualization:** Myriam Jaam, Lina Mohammad Naseralallah, Tarteel Ali Hussain, Shane Ashley Pawluk.

**Writing – original draft:** Myriam Jaam, Lina Mohammad Naseralallah, Tarteel Ali Hussain, Shane Ashley Pawluk.

**Writing – review & editing:** Myriam Jaam, Lina Mohammad Naseralallah, Tarteel Ali Hussain, Shane Ashley Pawluk.

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
