## [Decision Letter · Decision Letter 0]

27 Apr 2021

PONE-D-21-05743

Pharmacist-led educational interventions provided to healthcare providers to reduce medication errors: A systematic review and meta-analysis

PLOS ONE

Dear Dr. Jaam,

Thank you for submitting your manuscript to PLOS ONE. After careful consideration, we feel that it has merit but does not fully meet PLOS ONE’s publication criteria as it currently stands. Therefore, we invite you to submit a revised version of the manuscript that addresses the points raised during the review process.

We look forward to receiving your revised manuscript.

Kind regards,

Prof, Mojtaba Vaismoradi, PhD, MScN, BScN

Academic Editor

PLOS ONE

Reviewers' comments:

Reviewer #1: REVIEWER’S COMMENTS

Pharmacist-led educational interventions provided to healthcare providers to reduce medication errors: A systematic review and meta-analysis

PONE-D-21-05743

GENERAL COMMENTS

The topic is highly relevant. To a large extent, the report followed the Preferred Reporting Items for Systematic revies and Meta-Analyses (PRISMA) Checklist. However, the manuscript needs significant editing for language and writing quality, to eliminate ambiguities. This would improve the fluidity and readability of the text. The authors should provide more explanations to make the Discussion section clearer. Readers might appreciate this section better, if comparisons/contrasts are made between their findings and those from similar studies while bringing to the fore the relevance in pharmacy practice. The manuscript could be accepted for publication if the necessary corrections are made in the revised submission.

ABSTRACT

1. Line 53: Write the full meaning of PROSPERO with the abbreviation in bracket. End the sentence with a full stop.

2. Line 54: … twelve (12) studies … (It is usually better not to start a sentence with figures)

3. Line 65: Preferably, arrange the keywords in alphabetical order separated by semi-colons.

4. General: Abstract is clear and gives a good description of the study.

INTRODUCTION

1. Line 70: Medication errors are defined …

2.Line 81: “safety” was mentioned twice

3. Lines 85 & 95: You can rephrase the statements without including the authors’ names.

4. Line 90: … have been reported to increase …

5 Line 91: … at the same time, reducing the …

6. Line 98: … aimed to …

MATERIALS AND METHODS

1. Line 103: The authors need to write the full meaning of PRISMA with the abbreviation in brackets. The authors should cite one of the original publications of the PRISMA Statement or PRISMA Explanation and Elaboration.

2. Line 104: Write the full meaning of PROSPERO with the abbreviation in brackets (In addition, correct the spelling of your abbreviation). The authors need to provide citation or where the review protocol can be accessed.

3. Line 112: … two researchers, independently, to ensure …

4. Line 114: … two authors, independently OR two independent authors.

5. Line 115: Category C – Did the authors mean to write “Reduce*”?

6. Lines 118 & 119: For clarity of the inclusion criteria, it needs to be rephrased. The authors can opt for the use of lower-case Roman numerals.

7. Line 120: It is unnecessary to include “regardless of study design” as this creates ambiguity.

8. Lines 121 – 124: For clarity of the exclusion criteria, the authors can use lower-case Roman numerals for the listing.

9. Line 123: … led by students …

10 Line 131: … two independent researchers …

11. Line 133: Include citation for the Crowe Critical Appraisal Tool (CCAT)

12. Line 135: Each domain has a maximum score of five … (Write single digits as words)

13. Line 137: … two independent researchers …

14. Line 137/8: The CCAT was previously …

RESULTS

1. Line 149: … (n = 6857), the remaining articles …

2. Line 150: … two researchers. A total of 12 articles …

3. Line 166: Additionally, …

4. Line 167: Delete “or”

5. Table 2: The contents for the “medication error rate post-intervention” column is confusing. What do the figures after the duration in round brackets represent? Is this captured in the heading?

6. General: Be uniform in your report of the percentages and figures. Use either one decimal place or two decimal places.

Be consistent with how you write post or pre-intervention. Is it post-intervention or postintervention? Be consistent.

“et al.” should be italicized.

7. Line 176: CCAT appraised …

8. Line 183/4: … that varied in content …

9. Line 188: … two weeks …

10. Line 191: … six studies …

11. Line 204: … that summarized …

12. Line 214: The last statement needs to be rephrased. Starting a statement with “Nor” could confuse readers.

13. Line 218: Do you mean the ten studies presented that P-value of 0.0004? Was this stated in the studies or calculated by the authors? If any of the P-values were by the authors, then it should reflect in the Methods section (data analysis and synthesis). The level of significance should be included.

DISCUSSION

1. General: The Discussion section needs to be improved. The authors seemed to focus on making comparisons between the studies that were included in their study. This was similar to Results summary. A more robust Discussion should provide possible reasons for the findings; compare/contrast the findings with other studies; relate the findings to pharmacy practice.

2. Line 232 – 234: This should be in the Introduction section, as it is a justification for conducting the study. It should not appear in the Discussion section. The introductory paragraph of the Discussion should be a summary of the findings without repeating the results.

3. Line 237/8: Do not report figures in your Discussion section, as they should already be in the Results section.

4. Line 245: one month or 1-month

5. Line 267 – 269: Sentence should be rephrased.

6. Line 273 – 274: Sentence should be rephrased. It appears incomplete.

7. General: The use of different referencing styles should be corrected all through the manuscript.

8. Line 281 – 285: Sentence is complex and can be broken into simple sentences. For clarity, it is better to have two simple sentences than have one extremely long sentence with multiple punctuation marks and conjunctions.

9. Line 287: … In lieu, …

10. Line 289: … site-specific, based …

11. Line 291/2: The sentence should be rephrased.

12. Line 294: delete “All in all”.

13. Line 294: … the included studies …

14. Line 295: … CCAT …

15. Line 301: Write the full meaning of EQUATOR with the abbreviation in bracket.

16. Line 309: … was reported. (Delete “and was expected”)

17: General: The limitations were well described.

18. Line 340: Do you mean “To err is human”?

CONCLUSION

1. Line 348/9: It is not necessary to include the overall odds ratio in the Conclusion.

2. Line 350: … that summarize …

CONTRIBUTIONS

Line 365 – 368: All authors should be mentioned. All authors should be represented by their full abbreviations.

REFERENCES

1. Please follow the authors’ guidelines. There should be uniformity in the References. For instance, some Journal titles are abbreviated while others are not e.g., Reference 4, 11, 42. Check all.

2. Reference 40, 46: Include date cited.

FIGURES

1. Figure 1: The descriptions on the far-left are upside down.

2. Figure 1: The calculation is confusing as (9346 + 8) – 6857 = 2497 (The authors’ flow diagram shows 2498). The first sentence in the Results section confirms that 9354 articles were identified from the different databases and reference screening. The correct values should be inputted in the flow diagram.

Reviewer #2: Dear authors

I have several suggestions for improving your manuscript.

Abstract

1.Please write the sampling method, study design, and study time.

Introduction

2. I think it is better to remove the following sentence and add it to the end of the introduction before stating the purpose.

" This study sought to implement a remote corporate-based pharmacist into a home-based primary care practice to facilitate comprehensive medication reviews."

3. Please provide the references in the following sentences.

Line:79, 97,

4. At the end of line 175, change "a American…" to " an American….".

Methods

5. Please write the sampling method, study design, and study time.

6. Please explain inclusion and exclusion criteria.

7. Please write ethical considerations.

Results

8. In Table 1, the total percentage in the payer's column is 99.9%, please correct.

9. In table 1, the term Std should be clarified.

10. Please use the full form of "VTE" first used in table 3.

Discussion

11. You can suggest conducting experimental studies such as Randomised Clinical Trial (RCT) for future studies.

Best regards

---

## [Author Response · Author response to Decision Letter 0]

2 May 2021

Dear Madam/Sir,

Revision of Manuscript PONE-D-21-05743: Pharmacist-led educational interventions provided to healthcare providers to reduce medication errors: A systematic review and meta-analysis

The authors of the above-named manuscript would like to thank you for your e-mail dated 27 April 2021, containing the constructive comments of the reviewer. The authors very much appreciate the comments and have revised the manuscript accordingly as attached. We are positive that the comments helped in enhancing the quality and scientific merit of the paper.

Within the submission system we uploaded a document "Response to reviewers" which includes our response in details. 

All in all, the article was reviewed for language and writing with adjustments made throughout. Additionally, the discussion was adjusted to accommodate the reviewer’s comment which we believe adds value to the manuscript. 

We also wish to clarify in response to this comment: 5. Line 115: Category C – Did the authors mean to write “Reduce*”?

The authors used * to capture the following: Reduction, Reducing, Reduce. This was not a typo. 

Everything else was fully addressed including comments received from Reviewer #2. 

Thank you to the editor and the reviewer for the time spent on this manuscript. We look forward to receiving your kind response.

Sincerely yours,

The Corresponding Author

---

## [Decision Letter · Decision Letter 1]

26 May 2021

PONE-D-21-05743R1

Pharmacist-led educational interventions provided to healthcare providers to reduce medication errors: A systematic review and meta-analysis

PLOS ONE

Dear Dr. Jaam,

Thank you for submitting your manuscript to PLOS ONE. After careful consideration, we feel that it has merit but does not fully meet PLOS ONE’s publication criteria as it currently stands. Therefore, we invite you to submit a revised version of the manuscript that addresses the points raised during the review process.

We look forward to receiving your revised manuscript.

Kind regards,

Prof Mojtaba Vaismoradi, PhD, MScN, BScN

Academic Editor

PLOS ONE

Journal Requirements:

Reviewers' comments:

Reviewer #1: Review Comments to the Author

Pharmacist-led educational interventions provided to healthcare providers to reduce medication errors: A systematic review and meta-analysis

PONE-D-21-05743R1

GENERAL COMMENTS

The revised manuscript addressed the concerns in the first revision. The manuscript should be accepted for publication after the few corrections, stated below, are effected.

MATERIALS AND METHODS

Lines 103: meta-analyses (more appropriate in its plural form)

DISCUSSION

1. Line 228: et al.,

2. Line 303: Why is this line empty?

CONTRIBUTIONS

Line 346: Contributions

REFERENCES

Include date cited for accessed websites in References 40 & 46.

Reviewer #2: Dear authors

Thank you for your high-quality work. I have some minor comments for improving your manuscript.

1. In line 60 you stated " Studies reporting...." and mentioned only one reference, I checked the reference. Considering that the article is a systematic review study, it is better instead of " Studies reporting... " write" the results of a systematic review .... ". Because we are expected to refer to a number of articles when we mention the word "studies".

2.The introduction is well organized and follows a logical course. However, I suggest that it is better to mention the statistics of medication errors in the introduction section (about two or three sentences).

3.Please check the numbers in Table 3 again. The Overall Out of 40 of Fawaz et al study [21] is 23.5%, which is calculated in the table 25.5%.

4.Please correct the following:

Line 22: "increased" to " an increased or the increased " be corrected.

In line 143, the means of term "hospital" in the sentence "three covered emergency/acute care hospital" Is it hospital wards or hospitals?

Best regards

---

## [Author Response · Author response to Decision Letter 1]

30 May 2021

Response to reviewer 1: The authors thank the reviewer for their comments. The authors addressed all comments mentioned. 

Response to reviewer 2: The authors appreciate the comments provided. All comments were addressed in the revised manuscript. More statistics relevant to the topic were added in the introduction. All numbers were reviewed within table 3 to ensure accuracy. Modification was made to correct the error. All other comments were fully addressed.

---

## [Editor Report · Decision Letter 2]

9 Jun 2021

Pharmacist-led educational interventions provided to healthcare providers to reduce medication errors: A systematic review and meta-analysis

PONE-D-21-05743R2

Dear Dr. Jaam,

We’re pleased to inform you that your manuscript has been judged scientifically suitable for publication and will be formally accepted for publication once it meets all outstanding technical requirements.

Kind regards,

Prof, Mojtaba Vaismoradi, PhD, MScN, BScN

Academic Editor

PLOS ONE
---

## [Editor Report · Acceptance letter]

15 Jun 2021

PONE-D-21-05743R2 

Pharmacist-led educational interventions provided to healthcare providers to reduce medication errors: A systematic review and meta-analysis 

Dear Dr. Jaam:

I'm pleased to inform you that your manuscript has been deemed suitable for publication in PLOS ONE. Congratulations! Your manuscript is now with our production department. 

Kind regards, 

on behalf of

Professor Mojtaba Vaismoradi 

Academic Editor

PLOS ONE